# PointNet++: Deep Hierarchical Feature Learning on Point Sets in a Metric Space

**Charles R. Qi    Li Yi    Hao Su    Leonidas J. Guibas**
Stanford University

## Abstract

Few prior works study deep learning on point sets. PointNet [20] is a pioneer in this direction. However, by design PointNet does not capture local structures induced by the metric space points live in, limiting its ability to recognize fine-grained patterns and generalizability to complex scenes. In this work, we introduce a hierarchical neural network that applies PointNet recursively on a nested partitioning of the input point set. By exploiting metric space distances, our network is able to learn local features with increasing contextual scales. With further observation that point sets are usually sampled with varying densities, which results in greatly decreased performance for networks trained on uniform densities, we propose novel set learning layers to adaptively combine features from multiple scales. Experiments show that our network called PointNet++ is able to learn deep point set features efficiently and robustly. In particular, results significantly better than state-of-the-art have been obtained on challenging benchmarks of 3D point clouds.

## 1   Introduction

We are interested in analyzing geometric point sets which are collections of points in a Euclidean space. A particularly important type of geometric point set is point cloud captured by 3D scanners, e.g., from appropriately equipped autonomous vehicles. As a set, such data has to be invariant to permutations of its members. In addition, the distance metric defines local neighborhoods that may exhibit different properties. For example, the density and other attributes of points may not be uniform across different locations — in 3D scanning the density variability can come from perspective effects, radial density variations, motion, etc.

Few prior works study deep learning on point sets. PointNet [20] is a pioneering effort that directly processes point sets. The basic idea of PointNet is to learn a spatial encoding of each point and then aggregate all individual point features to a global point cloud signature. By its design, PointNet does not capture local structure induced by the metric. However, exploiting local structure has proven to be important for the success of convolutional architectures. A CNN takes data defined on regular grids as the input and is able to progressively capture features at increasingly larger scales along a multi-resolution hierarchy. At lower levels neurons have smaller receptive fields whereas at higher levels they have larger receptive fields. The ability to abstract local patterns along the hierarchy allows better generalizability to unseen cases.

We introduce a hierarchical neural network, named as PointNet++, to process a set of points sampled in a metric space in a hierarchical fashion. The general idea of PointNet++ is simple. We first partition the set of points into overlapping local regions by the distance metric of the underlying space. Similar to CNNs, we extract local features capturing fine geometric structures from small neighborhoods; such local features are further grouped into larger units and processed to produce higher level features. This process is repeated until we obtain the features of the whole point set.

The design of PointNet++ has to address two issues: how to generate the partitioning of the point set, and how to abstract sets of points or local features through a local feature learner. The two issues are correlated because the partitioning of the point set has to produce common structures across partitions, so that weights of local feature learners can be shared, as in the convolutional setting. We choose our local feature learner to be PointNet. As demonstrated in that work, PointNet is an effective architecture to process an unordered set of points for semantic feature extraction. In addition, this architecture is robust to input data corruption. As a basic building block, PointNet abstracts sets of local points or features into higher level representations. In this view, PointNet++ applies PointNet recursively on a nested partitioning of the input set.

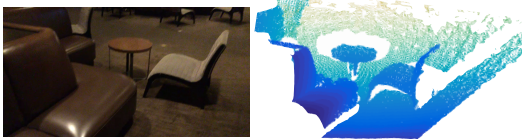

One issue that still remains is how to generate overlapping partitioning of a point set. Each partition is defined as a neighborhood ball in the underlying Euclidean space, whose parameters include centroid location and scale. To evenly cover the whole set, the centroids are selected among input point set by a farthest point sampling (FPS) algorithm. Compared with volumetric CNNs that scan the space with fixed strides, our local receptive fields are dependent on both the input data and the metric, and thus more efficient and effective.

Figure 1: Visualization of a scan captured from a Structure Sensor (left: RGB; right: point cloud).

Deciding the appropriate scale of local neighborhood balls, however, is a more challenging yet intriguing problem, due to the entanglement of feature scale and non-uniformity of input point set. We assume that the input point set may have variable density at different areas, which is quite common in real data such as Structure Sensor scanning [18] (see Fig. 1). Our input point set is thus very different from CNN inputs which can be viewed as data defined on regular grids with uniform constant density. In CNNs, the counterpart to local partition scale is the size of kernels. [25] shows that using smaller kernels helps to improve the ability of CNNs. Our experiments on point set data, however, give counter evidence to this rule. Small neighborhood may consist of too few points due to sampling deficiency, which might be insufficient to allow PointNets to capture patterns robustly.

A significant contribution of our paper is that PointNet++ leverages neighborhoods at multiple scales to achieve both robustness and detail capture. Assisted with random input dropout during training, the network learns to adaptively weight patterns detected at different scales and combine multi-scale features according to the input data. Experiments show that our PointNet++ is able to process point sets efficiently and robustly. In particular, results that are significantly better than state-of-the-art have been obtained on challenging benchmarks of 3D point clouds.

## 2 Problem Statement

Suppose that $\mathcal{X} = (M, d)$ is a discrete metric space whose metric is inherited from a Euclidean space $\mathbb{R}^n$, where $M \subseteq \mathbb{R}^n$ is the set of points and $d$ is the distance metric. In addition, the density of $M$ in the ambient Euclidean space may not be uniform everywhere. We are interested in learning set functions $f$ that take such $\mathcal{X}$ as the input (along with additional features for each point) and produce information of semantic interest regrading $\mathcal{X}$. In practice, such $f$ can be classification function that assigns a label to $\mathcal{X}$ or a segmentation function that assigns a per point label to each member of $M$.

## 3 Method

Our work can be viewed as an extension of PointNet [20] with added hierarchical structure. We first review PointNet (Sec. 3.1) and then introduce a basic extension of PointNet with hierarchical structure (Sec. 3.2). Finally, we propose our PointNet++ that is able to robustly learn features even in non-uniformly sampled point sets (Sec. 3.3).

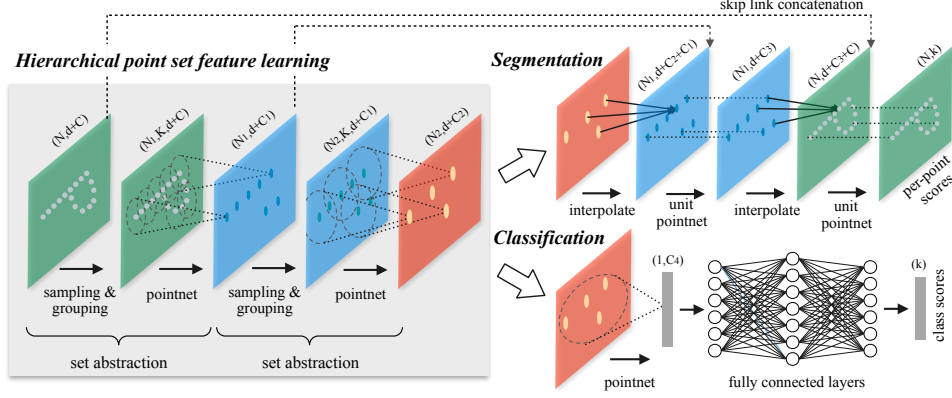

Figure 2: Illustration of our hierarchical feature learning architecture and its application for set segmentation and classification using points in 2D Euclidean space as an example. Single scale point grouping is visualized here. For details on density adaptive grouping, see Fig. 3

## 3.1 Review of PointNet [20]: A Universal Continuous Set Function Approximator

Given an unordered point set $\{x_1, x_2, ..., x_n\}$ with $x_i \in \mathbb{R}^d$, one can define a set function $f : \mathcal{X} \to \mathbb{R}$ that maps a set of points to a vector:

$$f(x_1, x_2, ..., x_n) = \gamma \left( \underset{i=1,...,n}{\text{MAX}} \{h(x_i)\} \right) \qquad (1)$$

where $\gamma$ and $h$ are usually multi-layer perceptron (MLP) networks.

The set function $f$ in Eq. 1 is invariant to input point permutations and can arbitrarily approximate any continuous set function [20]. Note that the response of $h$ can be interpreted as the spatial encoding of a point (see [20] for details).

PointNet achieved impressive performance on a few benchmarks. However, it lacks the ability to capture local context at different scales. We will introduce a hierarchical feature learning framework in the next section to resolve the limitation.

## 3.2 Hierarchical Point Set Feature Learning

While PointNet uses a single max pooling operation to aggregate the whole point set, our new architecture builds a hierarchical grouping of points and progressively abstract larger and larger local regions along the hierarchy.

Our hierarchical structure is composed by a number of *set abstraction* levels (Fig. 2). At each level, a set of points is processed and abstracted to produce a new set with fewer elements. The set abstraction level is made of three key layers: *Sampling layer*, *Grouping layer* and *PointNet layer*. The *Sampling layer* selects a set of points from input points, which defines the centroids of local regions. *Grouping layer* then constructs local region sets by finding "neighboring" points around the centroids. *PointNet layer* uses a mini-PointNet to encode local region patterns into feature vectors.

A set abstraction level takes an $N \times (d + C)$ matrix as input that is from $N$ points with $d$-dim coordinates and $C$-dim point feature. It outputs an $N' \times (d + C')$ matrix of $N'$ subsampled points with $d$-dim coordinates and new $C'$-dim feature vectors summarizing local context. We introduce the layers of a set abstraction level in the following paragraphs.

**Sampling layer.** Given input points $\{x_1, x_2, ..., x_n\}$, we use iterative farthest point sampling (FPS) to choose a subset of points $\{x_{i_1}, x_{i_2}, ..., x_{i_m}\}$, such that $x_{i_j}$ is the most distant point (in metric distance) from the set $\{x_{i_1}, x_{i_2}, ..., x_{i_{j-1}}\}$ with regard to the rest points. Compared with random sampling, it has better coverage of the entire point set given the same number of centroids. In contrast to CNNs that scan the vector space agnostic of data distribution, our sampling strategy generates receptive fields in a data dependent manner.

**Grouping layer.** The input to this layer is a point set of size $N \times (d + C)$ and the coordinates of a set of centroids of size $N' \times d$. The output are groups of point sets of size $N' \times K \times (d + C)$, where each group corresponds to a local region and $K$ is the number of points in the neighborhood of centroid points. Note that $K$ varies across groups but the succeeding *PointNet layer* is able to convert flexible number of points into a fixed length local region feature vector.

In convolutional neural networks, a local region of a pixel consists of pixels with array indices within certain Manhattan distance (kernel size) of the pixel. In a point set sampled from a metric space, the neighborhood of a point is defined by metric distance.

Ball query finds all points that are within a radius to the query point (an upper limit of $K$ is set in implementation). An alternative range query is $K$ nearest neighbor (kNN) search which finds a fixed number of neighboring points. Compared with kNN, ball query's local neighborhood guarantees a fixed region scale thus making local region feature more generalizable across space, which is preferred for tasks requiring local pattern recognition (e.g. semantic point labeling).

**PointNet layer.** In this layer, the input are $N'$ local regions of points with data size $N' \times K \times (d + C)$. Each local region in the output is abstracted by its centroid and local feature that encodes the centroid's neighborhood. Output data size is $N' \times (d + C')$.

The coordinates of points in a local region are firstly translated into a local frame relative to the centroid point: $x_i^{(j)} = x_i^{(j)} - \hat{x}^{(j)}$ for $i = 1, 2, ..., K$ and $j = 1, 2, ..., d$ where $\hat{x}$ is the coordinate of the centroid. We use PointNet [20] as described in Sec. 3.1 as the basic building block for local pattern learning. By using relative coordinates together with point features we can capture point-to-point relations in the local region.

## 3.3 Robust Feature Learning under Non-Uniform Sampling Density

As discussed earlier, it is common that a point set comes with non-uniform density in different areas. Such non-uniformity introduces a significant challenge for point set feature learning. Features learned in dense data may not generalize to sparsely sampled regions. Consequently, models trained for sparse point cloud may not recognize fine-grained local structures.

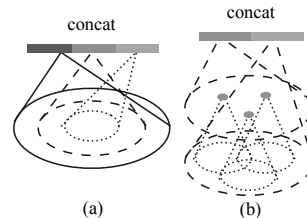

Ideally, we want to inspect as closely as possible into a point set to capture finest details in densely sampled regions. However, such close inspect is prohibited at low density areas because local patterns may be corrupted by the sampling deficiency. In this case, we should look for larger scale patterns in greater vicinity. To achieve this goal we propose density adaptive PointNet layers (Fig. 3) that learn to combine features from regions of different scales when the input

Figure 3: (a) Multi-scale grouping (MSG); (b) Multi-resolution grouping (MRG).

sampling density changes. We call our hierarchical network with density adaptive PointNet layers as *PointNet++*.

Previously in Sec. 3.2, each abstraction level contains grouping and feature extraction of a single scale. In PointNet++, each abstraction level extracts multiple scales of local patterns and combine them intelligently according to local point densities. In terms of grouping local regions and combining features from different scales, we propose two types of density adaptive layers as listed below.

**Multi-scale grouping (MSG).** As shown in Fig. 3 (a), a simple but effective way to capture multi-scale patterns is to apply grouping layers with different scales followed by according PointNets to extract features of each scale. Features at different scales are concatenated to form a multi-scale feature.

We train the network to learn an optimized strategy to combine the multi-scale features. This is done by randomly dropping out input points with a randomized probability for each instance, which we call *random input dropout*. Specifically, for each training point set, we choose a dropout ratio $\theta$ uniformly sampled from $[0, p]$ where $p \leq 1$. For each point, we randomly drop a point with probability $\theta$. In practice we set $p = 0.95$ to avoid generating empty point sets. In doing so we present the network with training sets of various sparsity (induced by $\theta$) and varying uniformity (induced by randomness in dropout). During test, we keep all available points.

**Multi-resolution grouping (MRG).** The MSG approach above is computationally expensive since it runs local PointNet at large scale neighborhoods for every centroid point. In particular, since the number of centroid points is usually quite large at the lowest level, the time cost is significant.

Here we propose an alternative approach that avoids such expensive computation but still preserves the ability to adaptively aggregate information according to the distributional properties of points. In Fig. 3 (b), features of a region at some level $L_i$ is a concatenation of two vectors. One vector (left in figure) is obtained by summarizing the features at each subregion from the lower level $L_{i-1}$ using the set abstraction level. The other vector (right) is the feature that is obtained by directly processing all raw points in the local region using a single PointNet.

When the density of a local region is low, the first vector may be less reliable than the second vector, since the subregion in computing the first vector contains even sparser points and suffers more from sampling deficiency. In such a case, the second vector should be weighted higher. On the other hand, when the density of a local region is high, the first vector provides information of finer details since it possesses the ability to inspect at higher resolutions recursively in lower levels.

Compared with MSG, this method is computationally more efficient since we avoids the feature extraction in large scale neighborhoods at lowest levels.

## 3.4 Point Feature Propagation for Set Segmentation

In set abstraction layer, the original point set is subsampled. However in set segmentation task such as semantic point labeling, we want to obtain point features for *all* the original points. One solution is to always sample all points as centroids in all set abstraction levels, which however results in high computation cost. Another way is to propagate features from subsampled points to the original points.

We adopt a hierarchical propagation strategy with distance based interpolation and across level skip links (as shown in Fig. 2). In a *feature propagation* level, we propagate point features from $N_l \times (d + C)$ points to $N_{l-1}$ points where $N_{l-1}$ and $N_l$ (with $N_l \leq N_{l-1}$) are point set size of input and output of set abstraction level $l$. We achieve feature propagation by interpolating feature values $f$ of $N_l$ points at coordinates of the $N_{l-1}$ points. Among the many choices for interpolation, we use inverse distance weighted average based on $k$ nearest neighbors (as in Eq. 2, in default we use $p = 2$, $k = 3$). The interpolated features on $N_{l-1}$ points are then concatenated with skip linked point features from the set abstraction level. Then the concatenated features are passed through a "unit pointnet", which is similar to one-by-one convolution in CNNs. A few shared fully connected and ReLU layers are applied to update each point's feature vector. The process is repeated until we have propagated features to the original set of points.

$$f^{(j)}(x) = \frac{\sum_{i=1}^{k} w_i(x) f_i^{(j)}}{\sum_{i=1}^{k} w_i(x)} \quad \text{where} \quad w_i(x) = \frac{1}{d(x, x_i)^p}, \; j = 1, ..., C \tag{2}$$

## 4 Experiments

**Datasets** We evaluate on four datasets ranging from 2D objects (MNIST [11]), 3D objects (ModelNet40 [31] rigid object, SHREC15 [12] non-rigid object) to real 3D scenes (ScanNet [5]). Object classification is evaluated by accuracy. Semantic scene labeling is evaluated by average voxel classification accuracy following [5]. We list below the experiment setting for each dataset:

- MNIST: Images of handwritten digits with 60k training and 10k testing samples.

- ModelNet40: CAD models of 40 categories (mostly man-made). We use the official split with 9,843 shapes for training and 2,468 for testing.

- SHREC15: 1200 shapes from 50 categories. Each category contains 24 shapes which are mostly organic ones with various poses such as horses, cats, etc. We use five fold cross validation to acquire classification accuracy on this dataset.

- ScanNet: 1513 scanned and reconstructed indoor scenes. We follow the experiment setting in [5] and use 1201 scenes for training, 312 scenes for test.

| Method | Error rate (%) |
|---|---|
| Multi-layer perceptron [24] | 1.60 |
| LeNet5 [11] | 0.80 |
| Network in Network [13] | **0.47** |
| PointNet (vanilla) [20] | 1.30 |
| PointNet [20] | 0.78 |
| Ours | 0.51 |

Table 1: MNIST digit classification.

| Method | Input | Accuracy (%) |
|---|---|---|
| Subvolume [21] | vox | 89.2 |
| MVCNN [26] | img | 90.1 |
| PointNet (vanilla) [20] | pc | 87.2 |
| PointNet [20] | pc | 89.2 |
| Ours | pc | 90.7 |
| Ours (with normal) | pc | **91.9** |

Table 2: ModelNet40 shape classification.

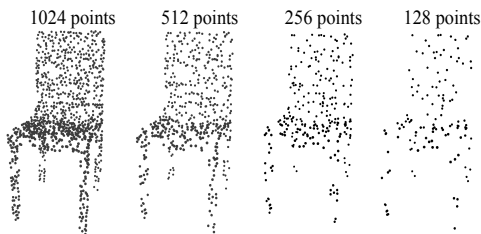

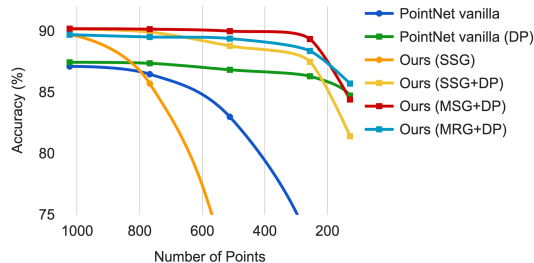

Figure 4: Left: Point cloud with random point dropout. Right: Curve showing advantage of our density adaptive strategy in dealing with non-uniform density. DP means random input dropout during training; otherwise training is on uniformly dense points. See Sec.3.3 for details.

## 4.1 Point Set Classification in Euclidean Metric Space

We evaluate our network on classifying point clouds sampled from both 2D (MNIST) and 3D (ModleNet40) Euclidean spaces. MNIST images are converted to 2D point clouds of digit pixel locations. 3D point clouds are sampled from mesh surfaces from ModelNet40 shapes. In default we use 512 points for MNIST and 1024 points for ModelNet40. In last row (ours normal) in Table 2, we use face normals as additional point features, where we also use more points ($N = 5000$) to further boost performance. All point sets are normalized to be zero mean and within a unit ball. We use a three-level hierarchical network with three fully connected layers [1]

**Results.** In Table 1 and Table 2, we compare our method with a representative set of previous state of the arts. Note that PointNet (vanilla) in Table 2 is the the version in [20] that does not use transformation networks, which is equivalent to our hierarchical net with only one level.

Firstly, our hierarchical learning architecture achieves significantly better performance than the non-hierarchical PointNet [20]. In MNIST, we see a relative 60.8% and 34.6% error rate reduction from PointNet (vanilla) and PointNet to our method. In ModelNet40 classification, we also see that using same input data size (1024 points) and features (coordinates only), ours is remarkably stronger than PointNet. Secondly, we observe that point set based method can even achieve better or similar performance as mature image CNNs. In MNIST, our method (based on 2D point set) is achieving an accuracy close to the Network in Network CNN. In ModelNet40, ours with normal information significantly outperforms previous state-of-the-art method MVCNN [26].

**Robustness to Sampling Density Variation.** Sensor data directly captured from real world usually suffers from severe irregular sampling issues (Fig. 1). Our approach selects point neighborhood of multiple scales and learns to balance the descriptiveness and robustness by properly weighting them.

We randomly drop points (see Fig. 4 left) during test time to validate our network's robustness to non-uniform and sparse data. In Fig. 4 right, we see MSG+DP (multi-scale grouping with random input dropout during training) and MRG+DP (multi-resolution grouping with random input dropout during training) are very robust to sampling density variation. MSG+DP performance drops by less than 1% from 1024 to 256 test points. Moreover, it achieves the best performance on almost all sampling densities compared with alternatives. PointNet vanilla [20] is fairly robust under density variation due to its focus on global abstraction rather than fine details. However loss of details also makes it less powerful compared to our approach. SSG (ablated PointNet++ with single scale grouping in each level) fails to generalize to sparse sampling density while SSG+DP amends the problem by randomly dropping out points in training time.

## 4.2 Point Set Segmentation for Semantic Scene Labeling

To validate that our approach is suitable for large scale point cloud analysis, we also evaluate on semantic scene labeling task. The goal is to predict semantic object label for points in indoor scans. [5] provides a baseline using fully convolutional neural network on voxelized scans. They purely rely on scanning geometry instead of RGB information and report the accuracy on a per-voxel basis. To make a fair comparison,

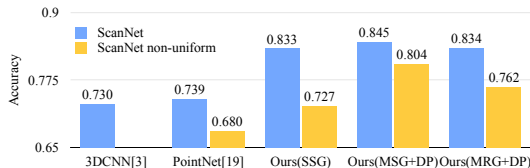

Figure 5: Scannet labeling accuracy.

we remove RGB information in all our experiments and convert point cloud label prediction into voxel labeling following [5]. We also compare with [20]. The accuracy is reported on a per-voxel basis in Fig. 5 (blue bar).

Our approach outperforms all the baseline methods by a large margin. In comparison with [5], which learns on voxelized scans, we directly learn on point clouds to avoid additional quantization error, and conduct data dependent sampling to allow more effective learning. Compared with [20], our approach introduces hierarchical feature learning and captures geometry features at different scales. This is very important for understanding scenes at multiple levels and labeling objects with various sizes. We visualize example scene labeling results in Fig. 6.

**Robustness to Sampling Density Variation**
To test how our trained model performs on scans with non-uniform sampling density, we synthesize virtual scans of Scannet scenes similar to that in Fig. 1 and evaluate our network on this data. We refer readers to supplementary material for how we generate the virtual scans. We evaluate our framework in three settings (SSG, MSG+DP, MRG+DP) and compare with a baseline approach [20].

Performance comparison is shown in Fig. 5 (yellow bar). We see that SSG performance greatly falls due to the sampling density shift from uniform point cloud to virtually scanned scenes. MRG network, on the other hand, is more robust to the sampling density shift since it is able to automatically switch to features depicting coarser

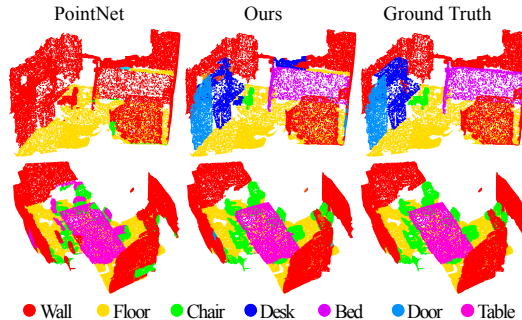

Figure 6: Scannet labeling results. [20] captures the overall layout of the room correctly but fails to discover the furniture. Our approach, in contrast, is much better at segmenting objects besides the room layout.

granularity when the sampling is sparse. Even though there is a domain gap between training data (uniform points with random dropout) and scanned data with non-uniform density, our MSG network is only slightly affected and achieves the best accuracy among methods in comparison. These prove the effectiveness of our density adaptive layer design.

## 4.3 Point Set Classification in Non-Euclidean Metric Space

In this section, we show generalizability of our approach to non-Euclidean space. In non-rigid shape classification (Fig. 7), a good classifier should be able to classify $(a)$ and $(c)$ in Fig. 7 correctly as the same category even given their difference in pose, which requires knowledge of intrinsic structure. Shapes in SHREC15 are 2D surfaces embedded in 3D space. Geodesic distances along the surfaces naturally induce a metric space. We show through experiments that adopting PointNet++ in this metric space is an effective way to capture intrinsic structure of the underlying point set.

For each shape in [12], we firstly construct the metric space induced by pairwise geodesic distances. We follow [23] to obtain an embedding metric that mimics geodesic distance. Next we extract intrinsic point features in this metric space including WKS [1], HKS [27] and multi-scale Gaussian curvature [16]. We use these features as input and then sample and group points according to the underlying metric space. In this way, our network learns to capture multi-scale intrinsic structure that is not influenced by the specific pose of a shape. Alternative design choices include using $XYZ$ coordinates as points feature or use Euclidean space $\mathbb{R}^3$ as the underlying metric space. We show below these are not optimal choices.

**Results.** We compare our methods with previous state-of-the-art method [14] in Table 3. [14] extracts geodesic moments as shape features and use a stacked sparse autoencoder to digest these features to predict shape category. Our approach using non-Euclidean metric space and intrinsic features achieves the best performance in all settings and outperforms [14] by a large margin.

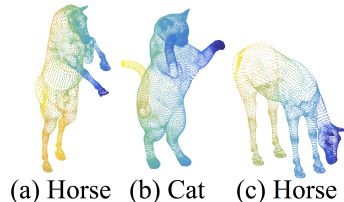

(a) Horse  (b) Cat  (c) Horse

Figure 7: An example of non-rigid shape classification.

Comparing the first and second setting of our approach, we see intrinsic features are very important for non-rigid shape classification. $XYZ$ feature fails to reveal intrinsic structures and is greatly influenced by pose variation. Comparing the second and third setting of our approach, we see using geodesic neighborhood is beneficial compared with Euclidean neighborhood. Euclidean neighborhood might include points far away on surfaces and this neighborhood could change dramatically when shape affords non-rigid deformation. This introduces difficulty for effective weight sharing since the local structure could become combinatorially complicated. Geodesic neighborhood on surfaces, on the other hand, gets rid of this issue and improves the learning effectiveness.

|          | Metric space   | Input feature      | Accuracy (%) |
|----------|----------------|--------------------|--------------|
| DeepGM [14] | -           | Intrinsic features | 93.03        |
|          | Euclidean      | XYZ                | 60.18        |
| Ours     | Euclidean      | Intrinsic features | 94.49        |
|          | Non-Euclidean  | Intrinsic features | **96.09**    |

Table 3: SHREC15 Non-rigid shape classification.

## 4.4   Feature Visualization.

In Fig. 8 we visualize what has been learned by the first level kernels of our hierarchical network. We created a voxel grid in space and aggregate local point sets that activate certain neurons the most in grid cells (highest 100 examples are used). Grid cells with high votes are kept and converted back to 3D point clouds, which represents the pattern that neuron recognizes. Since the model is trained on ModelNet40 which is mostly consisted of furniture, we see structures of planes, double planes, lines, corners etc. in the visualization.

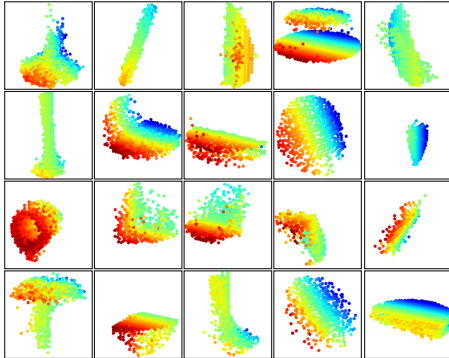

## 5   Related Work

The idea of hierarchical feature learning has been very successful. Among all the learning models, convolutional neural network [10; 25; 8] is one of the most prominent ones. However, convolution does not apply to unordered point sets with distance metrics, which is the focus of our work.

Figure 8: 3D point cloud patterns learned from the first layer kernels. The model is trained for ModelNet40 shape classification (20 out of the 128 kernels are randomly selected). Color indicates point depth (red is near, blue is far).

A few very recent works [20; 28] have studied how to apply deep learning to unordered sets. They ignore the underlying distance metric even if the point set does possess one. As a result, they are unable to capture local context of points and are sensitive to global set translation and normalization. In this work, we target at points sampled from a metric space and tackle these issues by explicitly considering the underlying distance metric in our design.

Point sampled from a metric space are usually noisy and with non-uniform sampling density. This affects effective point feature extraction and causes difficulty for learning. One of the key issue is to select proper scale for point feature design. Previously several approaches have been developed regarding this [19; 17; 2; 6; 7; 30] either in geometry processing community or photogrammetry and remote sensing community. In contrast to all these works, our approach learns to extract point features and balance multiple feature scales in an end-to-end fashion.

In 3D metric space, other than point set, there are several popular representations for deep learning, including volumetric grids [21; 22; 29], and geometric graphs [3; 15; 33]. However, in none of these works, the problem of non-uniform sampling density has been explicitly considered.

## 6   Conclusion

In this work, we propose PointNet++, a powerful neural network architecture for processing point sets sampled in a metric space. PointNet++ recursively functions on a nested partitioning of the input point set, and is effective in learning hierarchical features with respect to the distance metric. To handle the non uniform point sampling issue, we propose two novel set abstraction layers that intelligently aggregate multi-scale information according to local point densities. These contributions enable us to achieve state-of-the-art performance on challenging benchmarks of 3D point clouds.

In the future, it's worthwhile thinking how to accelerate inference speed of our proposed network especially for MSG and MRG layers by sharing more computation in each local regions. It's also interesting to find applications in higher dimensional metric spaces where CNN based method would be computationally unfeasible while our method can scale well.

**Acknowledgement.** The authors would like to acknowledge the support of a Samsung GRO grant, NSF grants IIS-1528025 and DMS-1546206, and ONR MURI grant N00014-13-1-0341.

## Footnotes

[1]See supplementary for more details on network architecture and experiment preparation.

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
