[Supplementary Material · NIPS17_CameraReady_Supp.pdf]

# PointNet++: Deep Hierarchical Feature Learning on Point Sets in a Metric Space
# Supplementary Material

**Charles R. Qi**    **Li Yi**    **Hao Su**    **Leonidas J. Guibas**
Stanford University

## A    Overview

This supplementary material provides more details on experiments in the main paper and includes more experiments to validate and analyze our proposed method.

In Sec B we provide specific network architectures used for experiments in the main paper and also describe details in data preparation and training. In Sec C we show more experimental results including benchmark performance on part segmentation and analysis on neighborhood query, sensitivity to sampling randomness and time space complexity.

## B    Details in Experiments

**Architecture protocol.**    We use following notations to describe our network architecture.

SA($K$,$r$,$[l_1, ..., l_d]$) is a set abstraction (SA) level with $K$ local regions of ball radius $r$ using PointNet of $d$ fully connected layers with width $l_i$ ($i = 1, ..., d$). SA($[l_1, ...l_d]$) is a global set abstraction level that converts set to a single vector. In multi-scale setting (as in MSG), we use SA($K$, $[r^{(1)}, ..., r^{(m)}]$, $[[l_1^{(1)}, ..., l_d^{(1)}], ..., [l_1^{(m)}, ..., l_d^{(m)}]]$) to represent MSG with $m$ scales.

FC($l$,$dp$) represents a fully connected layer with width $l$ and dropout ratio $dp$. FP($l_1, ..., l_d$) is a feature propagation (FP) level with $d$ fully connected layers. It is used for updating features concatenated from interpolation and skip link. All fully connected layers are followed by batch normalization and ReLU except for the last score prediction layer.

### B.1    Network Architectures

For all classification experiments we use the following architecture (Ours SSG) with different $K$ (number of categories):

$$SA(512, 0.2, [64, 64, 128]) \rightarrow SA(128, 0.4, [128, 128, 256]) \rightarrow SA([256, 512, 1024]) \rightarrow$$
$$FC(512, 0.5) \rightarrow FC(256, 0.5) \rightarrow FC(K)$$

The multi-scale grouping (MSG) network (PointNet++) architecture is as follows:

$$SA(512, [0.1, 0.2, 0.4], [[32, 32, 64], [64, 64, 128], [64, 96, 128]]) \rightarrow$$
$$SA(128, [0.2, 0.4, 0.8], [[64, 64, 128], [128, 128, 256], [128, 128, 256]]) \rightarrow$$
$$SA([256, 512, 1024]) \rightarrow FC(512, 0.5) \rightarrow FC(256, 0.5) \rightarrow FC(K)$$

The cross level multi-resolution grouping (MRG) network's architecture uses three branches:

$$\text{Branch 1: } SA(512, 0.2, [64, 64, 128]) \rightarrow SA(64, 0.4, [128, 128, 256])$$

Branch 2: $SA(512, 0.4, [64, 128, 256])$ using $r = 0.4$ regions of original points

Branch 3: $SA(64, 128, 256, 512)$ using all original points.

Branch 4: $SA(256, 512, 1024)$.

Branch 1 and branch 2 are concatenated and fed to branch 4. Output of branch 3 and branch4 are then concatenated and fed to $FC(512, 0.5) \rightarrow FC(256, 0.5) \rightarrow FC(K)$ for classification.

Network for semantic scene labeling (last two fully connected layers in FP are followed by dropout layers with drop ratio 0.5):

$$SA(1024, 0.1, [32, 32, 64]) \rightarrow SA(256, 0.2, [64, 64, 128]) \rightarrow$$
$$SA(64, 0.4, [128, 128, 256]) \rightarrow SA(16, 0.8, [256, 256, 512]) \rightarrow$$
$$FP(256, 256) \rightarrow FP(256, 256) \rightarrow FP(256, 128) \rightarrow FP(128, 128, 128, 128, K)$$

Network for semantic and part segmentation (last two fully connected layers in FP are followed by dropout layers with drop ratio 0.5):

$$SA(512, 0.2, [64, 64, 128]) \rightarrow SA(128, 0.4, [128, 128, 256]) \rightarrow SA([256, 512, 1024]) \rightarrow$$
$$FP(256, 256) \rightarrow FP(256, 128) \rightarrow FP(128, 128, 128, 128, K)$$

## B.2 Virtual Scan Generation

In this section, we describe how we generate labeled virtual scan with non-uniform sampling density from ScanNet scenes. For each scene in ScanNet, we set camera location $1.5m$ above the centroid of the floor plane and rotate the camera orientation in the horizontal plane evenly in $8$ directions. In each direction, we use a image plane with size $100px$ by $75px$ and cast rays from camera through each pixel to the scene. This gives a way to select visible points in the scene. We could then generate $8$ virtual scans for each test scene similar and an example is shown in Fig. 1. Notice point samples are denser in regions closer to the camera.

(a) ScanNet labeled scene     (b) ScanNet non-uniform

Figure 1: Virtual scan generated from ScanNet

## B.3 MNIST and ModelNet40 Experiment Details

For MNIST images, we firstly normalize all pixel intensities to range $[0, 1]$ and then select all pixels with intensities larger than 0.5 as valid digit pixels. Then we convert digit pixels in an image into a 2D point cloud with coordinates within $[-1, 1]$, where the image center is the origin point. Augmented points are created to add the point set up to a fixed cardinality (512 in our case). We jitter the initial point cloud (with random translation of Gaussian distribution $\mathcal{N}(0, 0.01)$ and clipped to 0.03) to generate the augmented points. For ModelNet40, we uniformly sample $N$ points from CAD models surfaces based on face area.

For all experiments, we use Adam [3] optimizer with learning rate 0.001 for training. For data augmentation, we randomly scale object, perturb the object location as well as point sample locations. We also follow [5] to randomly rotate objects for ModelNet40 data augmentation. We use TensorFlow and GTX 1080, Titan X for training. All layers are implemented in CUDA to run GPU. It takes around 20 hours to train our model to convergence.

## B.4 ScanNet Experiment Details

To generate training data from ScanNet scenes, we sample 1.5m by 1.5m by 3m cubes from the initial scene and then keep the cubes where $\geq 2\%$ of the voxels are occupied and $\geq 70\%$ of the surface voxels have valid annotations (this is the same set up in [2]). We sample such training cubes on the fly and random rotate it along the up-right axis. Augmented points are added to the point set to make a fixed cardinality (8192 in our case). During test time, we similarly split the test scene into smaller cubes and get label prediction for every point in the cubes first, then merge label prediction in all the cubes from a same scene. If a point get different labels from different cubes, we will just conduct a majority voting to get the final point label prediction.

## B.5 SHREC15 Experiment Details

We randomly sample 1024 points on each shape both for training and testing. To generate the input intrinsic features, we to extract 100 dimensional WKS, HKS and multiscale Gaussian curvature respectively, leading to a 300 dimensional feature vector for each point. Then we conduct PCA to reduce the feature dimension to 64. We use a 8 dimensional embedding following [6] to mimic the geodesic distance, which is used to describe our non-Euclidean metric space while choosing the point neighborhood.

# C   More Experiments

In this section we provide more experiment results to validate and analyze our proposed network architecture.

## C.1 Semantic Part Segmentation

Following the setting in [7], we evaluate our approach on part segmentation task assuming category label for each shape is already known. Taken shapes represented by point clouds as input, the task is to predict a part label for each point. The dataset contains 16,881 shapes from 16 classes, annotated with 50 parts in total. We use the official train test split following [1].

We equip each point with its normal direction to better depict the underlying shape. This way we could get rid of hand-crafted geometric features as is used in [7; 8]. We compare our framework with traditional learning based techniques [7], as well as state-of-the-art deep learning approaches [4; 8] in Table 1. Point intersection over union (IoU) is used as the evaluation metric, averaged across all part classes. Cross-entropy loss is minimized during training. On average, our approach achieves the best performance. In comparison with [4], our approach performs better on most of the categories, which proves the importance of hierarchical feature learning for detailed semantic understanding. Notice our approach could be viewed as implicitly building proximity graphs at different scales and operating on these graphs, thus is related to graph CNN approaches such as [8]. Thanks to the flexibility of our multi-scale neighborhood selection as well as the power of set operation units, we could achieve better performance compared with [8]. Notice our set operation unit is much simpler compared with graph convolution kernels, and we do not need to conduct expensive eigen decomposition as opposed to [8]. These make our approach more suitable for large scale point cloud analysis.

|  | mean | aero | bag | cap | car | chair | ear phone | guitar | knife | lamp | laptop | motor | mug | pistol | rocket | skate board | table |
|---|---|---|---|---|---|---|---|---|---|---|---|---|---|---|---|---|---|
| Yi [7] | 81.4 | 81.0 | 78.4 | 77.7 | 75.7 | 87.6 | 61.9 | 92.0 | 85.4 | 82.5 | 95.7 | 70.6 | 91.9 | 85.9 | 53.1 | 69.8 | 75.3 |
| PN [4] | 83.7 | 83.4 | 78.7 | 82.5 | 74.9 | 89.6 | 73.0 | 91.5 | 85.9 | 80.8 | 95.3 | 65.2 | 93.0 | 81.2 | 57.9 | 72.8 | 80.6 |
| SSCNN [8] | 84.7 | 81.6 | 81.7 | 81.9 | 75.2 | 90.2 | 74.9 | 93.0 | 86.1 | 84.7 | 95.6 | 66.7 | 92.7 | 81.6 | 60.6 | 82.9 | 82.1 |
| Ours | 85.1 | 82.4 | 79.0 | 87.7 | 77.3 | 90.8 | 71.8 | 91.0 | 85.9 | 83.7 | 95.3 | 71.6 | 94.1 | 81.3 | 58.7 | 76.4 | 82.6 |

Table 1: Segmentation results on ShapeNet part dataset.

## C.2 Neighborhood Query: kNN v.s. Ball Query.

Here we compare two options to select a local neighborhood. We used radius based ball query in our main paper. Here we also experiment with kNN based neighborhood search and also play with

different search radius and $k$. In this experiment all training and testing are on ModelNet40 shapes with uniform sampling density. 1024 points are used. As seen in Table 2, radius based ball query is slightly better than kNN based method. However, we speculate in very non-uniform point set, kNN based query will results in worse generalization ability. Also we observe that a slightly large radius is helpful for performance probably because it captures richer local patterns.

| kNN (k=16) | kNN (k=64) | radius (r=0.1) | radius (r=0.2) |
|:---:|:---:|:---:|:---:|
| 89.3 | 90.3 | 89.1 | 90.7 |

Table 2: Effects of neighborhood choices. Evaluation metric is classification accuracy (%) on ModelNet 40 test set.

### C.3 Effect of Randomness in Farthest Point Sampling.

For the *Sampling layer* in our set abstraction level, we use farthest point sampling (FPS) for point set sub sampling. However FPS algorithm is random and the subsampling depends on which point is selected first. Here we evaluate the sensitivity of our model to this randomness. In Table 3, we test our model trained on ModelNet40 for feature stability and classification stability.

To evaluate feature stability we extract global features of all test samples for 10 times with different random seed. Then we compute mean features for each shape across the 10 sampling. Then we compute standard deviation of the norms of feature's difference from the mean feature. At last we average all std. in all feature dimensions as reported in the table. Since features are normalized into 0 to 1 before processing, the 0.021 difference means a 2.1% deviation of feature norm.

For classification, we observe only a 0.17% standard deviation in test accuracy on all ModelNet40 test shapes, which is robust to sampling randomness.

| Feature difference std. | Accuracy std. |
|:---:|:---:|
| 0.021 | 0.0017 |

Table 3: Effects of randomness in FPS (using ModelNet40).

### C.4 Time and Space Complexity.

Table 4 summarizes comparisons of time and space cost between a few point set based deep learning method. We record forward time with a batch size 8 using TensorFlow 1.1 with a single GTX 1080. The first batch is neglected since there is some preparation for GPU. While PointNet (vanilla) [4] has the best time efficiency, our model without density adaptive layers achieved smallest model size with fair speed.

It's worth noting that ours MSG, while it has good performance in non-uniformly sampled data, it's 2x expensive than SSG version due the multi-scale region feature extraction. Compared with MSG, MRG is more efficient since it uses regions across layers.

| | PointNet (vanilla) | PointNet | Ours (SSG) | Ours (MSG) | Ours (MRG) |
|---|:---:|:---:|:---:|:---:|:---:|
| Model size (MB) | 9.4 | 40 | 8.7 | 12 | 24 |
| Forward time (ms) | 11.6 | 25.3 | 82.4 | 163.2 | 87.0 |

Table 4: Model size and inference time (forward pass) of several networks.