[Reviews · NeurIPS 2017]

Reviewer 1



This paper is improving the PointNet [19] with a recursive approach to introduce local context learning with gradually increasing scales. + The exposition is clear and the math is easy to follow. + Experiments show that PointNet++ achieves the best results on pointset benchmarks. - Even though the motivation builds upon local detail recovery ability of PointNet++, this is not strongly supported by the results. I am leaning towards acceptance due to the fact that clear experiments are provided on benchmarks with state of the art results.

Reviewer 2



This paper is a follow-up on the very recent PointNet paper. Its main contribution is to make PointNet more similar to CNNs by introducing the spatial relation between points (in the form of grouping), and thus a form of locality and shift invariance for the low level descriptors. While the contribution is straightforward to explain, and at the end not a huge step from PointNet, it is well presented, evaluated, , it was probably some work to make it work efficiently (and it would be good to add numbers about speed in the paper, I don't think I saw them and sampling, centring etc. may improve the time quite a lot) and the results are satisfying (even if they are not amazing: for classification, I would consider the most meaningful comparison pointnet vs. pointnet++ without normal, so 89.2/90.7 . For segmentation, the comparison without normals should be added ) As a sidenote, I appreciated the results on SHREC non rigid shape classification, I don't know much paper evaluating both for it and ModelNet.

Reviewer 3



As clearly indicated in the title, this paper submission is an extension of the PointNet work of [19], to appear at CVPR 2017. The goal is to classify and segment (3D) point clouds. Novel contributions over [19] are the use of a hierarchical network, leveraging neighbourhoods at different scales, and a mechanism to deal with varying sampling densities, effectively generating receptive fields that vary in a data dependent manner. All this leads to state-of-the-art results. PointNet++ seems an important extension over PointNet, in that it allows to properly exploit local spatial information. Yet the impact on the overall performance is just 1-2 percent. Some more experimental or theoretical analysis would have been appreciated. For instance: - A number of alternative sampling options, apart from most distant point, are mentioned in the text, but not compared against in the experiments. - In the conclusions, it is mentioned that MSG is more robust than MRG but worse in terms of computational efficiency. I'd like to see such a claim validated by a more in-depth analysis of the computational load, or at least some indicative numbers.